# Development of a Prognostic Model of Overall Survival for Metastatic Hormone-Naïve Prostate Cancer in Japanese Men

**DOI:** 10.3390/cancers14194822

**Published:** 2022-10-02

**Authors:** Ryunosuke Nakagawa, Hiroaki Iwamoto, Tomoyuki Makino, Renato Naito, Suguru Kadomoto, Norihito Akatani, Hiroshi Yaegashi, Shohei Kawaguchi, Takahiro Nohara, Kazuyoshi Shigehara, Kouji Izumi, Yoshifumi Kadono, Atsushi Takamatsu, Kotaro Yoshida, Atsushi Mizokami

**Affiliations:** 1Department of Integrative Cancer Therapy and Urology, Kanazawa University Graduate School of Medical Science, Kanazawa 920-8641, Japan; 2Department of Urology, Ishikawa Prefectural Central Hospital, Kanazawa 920-8530, Japan; 3Department of Nuclear Medicine, Kanazawa University Hospital, Kanazawa 920-8641, Japan; 4Department of Radiology, Kanazawa University Graduate School of Medical Sciences, Kanazawa 920-0934, Japan

**Keywords:** hormone-naïve prostate cancer, prognostic model, Gleason pattern, bone scan index, lactate dehydrogenase

## Abstract

**Simple Summary:**

Treatment strategies have changed dramatically in recent years with the development of a variety of agents for metastatic hormone-naïve prostate cancer. There is a need to identify prognostic factors for the appropriate choice of treatment for patients with hormone-naïve prostate cancer in Japanese men. Among the prostate cancer patients receiving treatment at our institution from 2000 to 2019, 198 patients with bone or visceral metastases at the initial diagnosis were included in the study. We retrospectively examined these factors of the overall survival, and identified Gleason pattern 5 content, bone scan index ≥ 1.5, and lactate dehydrogenase evels ≥ 300 IU/L as prognostic factors. Using these three factors, we developed a new prognostic model for overall survival that can more objectively predict the prognosis of patients simply and objectively.

**Abstract:**

Background: Treatment strategies have changed dramatically in recent years with the development of a variety of agents for metastatic hormone-naïve prostate cancer (mHNPC). There is a need to identify prognostic factors for the appropriate choice of treatment for patients with mHNPC, and we retrospectively examined these factors. Methods: Patients with mHNPC treated at our institution from 2000 to 2019 were included in this study. Overall survival (OS) was estimated retrospectively using the Kaplan–Meier method, and factors associated with OS were identified using univariate and multivariate analyses. A prognostic model was then developed based on the factors identified. Follow-up was terminated on 24 October 2021. Results: The median follow-up duration was 44.2 months, whereas the median OS was 85.2 months, with 88 patients succumbing to their disease. Multivariate analysis identified Gleason pattern (GP) 5 content, bone scan index (BSI) ≥ 1.5, and lactate dehydrogenase (LDH) levels ≥ 300 IU/L as prognostic factors associated with OS. We also developed a prognostic model that classified patients with mHNPC as low risk with no factor, intermediate risk with one factor, and high risk with two or three factors. Conclusions: Three prognostic factors for OS were identified in patients with mHNPC, namely GP5 inclusion, BSI ≥ 1.5, and LDH ≥ 300. Using these three factors, we developed a new prognostic model for OS that can more objectively predict patient prognosis.

## 1. Introduction

Prostate cancer (PC) is the most common malignancy in men and a leading cause of cancer-related deaths in developed countries [1]. Approximately 10–20% of patients have de novo metastatic disease, with the number of patients diagnosed with metastatic PC only increasing [2]. Since 1940, the standard treatment for metastatic PC has been androgen deprivation therapy (ADT) Ref [3]. While newly diagnosed metastatic PC initially responds to ADT, it can become resistant and progress to castration-resistant PC (CRPC). Despite available treatments for CRPC, including alternative ADT [4], androgen receptor axis targeted agents (ARAT), chemotherapy [5,6], and radium-223 [7], the disease is often fatal in the end. However, six randomized trials found that the combination of drugs, such as docetaxel, abiraterone acetate, enzalutamide, and apalutamide, improved outcomes for patients with metastatic hormone-naïve PC (mHNPC) compared to ADT alone [8,9,10,11,12,13,14]. In Japan, abiraterone, enzalutamide, and apalutamide, have been approved for the upfront treatment of mHNPC, and have become treatment options. These reports have increased the number of treatment options for mHNPC. Interestingly, the survival benefits of these novel therapies vary depending on the extent of metastasis and severity of the cancer. The LATITUDE trial showed that the combination of abiraterone acetate and prednisolone with ADT prompted longer overall survival (OS) compared to ADT alone in high-risk mHNPC patients, although the therapeutic benefits expected in low-risk mHNPC patients remains unknown [11]. In the CHAARTED trial, the combination of prior chemotherapy with docetaxel and ADT significantly prolonged the survival of mHNPC patients; however, their subgroup analysis showed that the low-volume group exhibited no improvement in OS [10]. In other words, a certain number of patients can be expected to survive for a long time with ADT alone, and overtreatment can be minimized by identifying prognostic factors. Roy et al. created a nomogram of mHNPC patients treated with upfront ARAT [15]. Their nomogram is based on data from high-risk patients enrolled in the LATITUDE trial, and is a very effective tool because it is composed of items that are used in daily clinical practice. In this study, we retrospectively examined prognostic factors to develop a prognostic model for mHNPC patients in Japan.

## 2. Materials and Methods

### 2.1. Patient Selection

Among the PC patients receiving treatment at Kanazawa University Hospital from 2000 to 2019, 198 patients with bone or visceral metastases at the initial diagnosis were included in the study. All patients were pathologically diagnosed PC, and distant metastasis was detected through computed tomography and/or bone scans performed at the time of diagnosis.

### 2.2. Collection of Clinical Data

Age, Gleason pattern (GP), prostate-specific antigen (PSA), bone scan index (BSI), metastasis location, C-reactive protein (CRP), neutrophil-to-lymphocyte ratio (NLR), hemoglobin (Hb), alkaline phosphatase (ALP), and lactate dehydrogenase (LDH), were obtained from medical records and retrospectively investigated and analyzed for factors associated with OS. OS was measured from the diagnosis of PC until death or last follow-up. Follow-up was terminated on 24 October 2021.

Clinical stage was determined based on the 8th edition of the Union for International Cancer Control Tumor, Node, Metastasis classification, published in 2017. The BSI was developed as a marker of the total amount of bone metastasis using whole-body scintigraphy with 99mTc-MDP, which was calculated using the BONENAVI version 2 software program (FUJIFILM Toyama Chemical Co., Ltd., Tokyo, Japan; Exini Bone, Exini Diagnostics, Lund, Sweden) and was used herein [16]. The BSI represents the percentage of total skeletal mass taken up by the tumor, and is a reproducible quantitative expression of tumor burden seen on bone. In addition, the probability of abnormality is calculated by detecting hyperaccumulated areas in the bone scintigraphic image, thus preventing missed areas and enabling the objective evaluation of bone metastasis [17].

### 2.3. Statistical Analyses

OS was estimated using the Kaplan–Meier method, with differences being compared using log-rank tests. We evaluated the predictive impact of several potential factors on the OS patients using the Cox proportional hazards model. Hazard ratio (HR) and 95% confidence intervals (CI) were calculated. Thereafter, a prognostic model was developed based on the identified factors. Statistical analyses were performed using the commercially available software Prism 8 (GraphPad, San Diego, CA, USA) and the SPSS ver. 25.0 (SPSS Inc., Chicago, IL, USA), with *p* values of <0.05 indicating statistical significance. Nomogram was created using the R statistical software, version 3.6.3 (R Foundation for Statistical Computing, Vienna, Austria).

### 2.4. Ethical Considerations

This study was approved by the institutional review board of Kanazawa University Hospital (2016-328). Informed consent was obtained in the form of opt-out posted at our facility allowed by Medical Ethics Committee of Kanazawa University. All methods were performed in accordance with relevant guidelines and regulations.

## 3. Results

### 3.1. Overall Survival of Patients Classified by LATITUDE and CHAARTED Criteria

The characteristics of the 198 patients with mHNPC with bone or visceral metastases are summarized in Table 1. The median age was 71 (65–78) years, with 58.1% having lymph node metastasis, 2.5% having M1a disease, 83.3% having M1b disease, and 14.1% having M1c disease. Visceral metastases were found in the lungs (23 patients, 11.6%), liver (3 patients, 1.5%), and adrenal gland (1 patient, 0.5%). The median initial PSA was 230.5 (72.7–859.35) ng/mL, with 54.0% having Gleason score (GS) ≥ 9. The initial treatment consisted of combined androgen therapy (CAB) or luteinizing hormone-releasing hormone agonists alone for most patients (93.4%).

The median follow-up duration and median OS were 44.2 and 85.2 months, respectively, and 88 patients have died. The high-risk group had a significantly shorter OS (HR: 2.45, 95% CI 1.48–4.00; *p* < 0.0001) than the low-risk group based on LATITUDE criteria (Figure 1a). The median OS was 135.0 and 55.06 months in the low- and high-risk groups, respectively. Moreover, the high-volume group had a significantly shorter OS (HR: 2.55, 95% CI 1.54–4.24; *p* < 0.0001) than the low-volume group based on the CHAARTED criteria (Figure 1b). The median OS was 135.0 and 52.93 months in the low- and high-volume groups, respectively.

### 3.2. Identification of Prognostic Factors in Overall Survival

Univariate analysis identified inclusion of GP 5 (HR: 2.78, 95% CI 1.72–4.47; *p* < 0.001), BSI ≥ 1.5 (HR: 1.91, 95% CI 1.03–3.53; *p* = 0.040), and LDH ≥ 300 IU/L (HR: 6.08, 95% CI 2.95–12.50; *p* < 0.001), as significant prognostic factors for OS, although age, PSA level, visceral metastasis, CRP, NLR, Hb, and ALP, were not in this cohort (Table 2). Although BSI was found to be a prognostic factor in patients with mHNPC, one other method for assessing bone metastases in prostate cancer is the extent of disease (EOD) score, proposed by Soloway et al. in 1988, and is a method for assessing bone metastasis in prostate cancer [18]. Univariate analysis of patients with low EOD scores (scores 1 and 2) and those with high EOD scores (scores 3 and 4) was performed, but it was not a significant predictor of prognosis (HR: 1.52, 95% CI 0.92–2.52; *p* = 0.11). OS was also investigated in Kaplan-Meier, but there was no significant difference between these two groups (HR: 1.49, 95% CI 0.90–2.49; *p* = 0.10) (Appendix A).

Multivariate analysis identified inclusion of GP 5 (HR 2.77, 95% CI 1.03–7.49; *p* = 0.045), BSI ≥ 1.5 (HR: 3.48, 95% CI 1.10–11.00; *p* = 0.033), and LDH ≥ 300 IU/L (HR: 8.11, 95% CI 1.99–33.11; *p* = 0.004), as factors associated with an increased risk of OS, similar to that in the univariate analysis. After comparing the three factors identified and detected as significant prognostic factors in both groups, OS was significantly shorter with GP 5 inclusion (HR: 2.58, 95% CI 1.72–4.47; *p* < 0.001) than with GP 5 exclusion (Figure 2a), in the BSI ≥ 1.5 group (HR: 3.23, 95% CI 1.77–5.89; *p* = 0.037) than in the BSI < 1.5 group (Figure 2b), and in the LDH ≥300 IU/L group (HR: 4.17, 95% CI 2.10–8.30; *p* < 0.0001) than in the LDH < 300 IU/L group (Figure 2c). Cancer-specific survival (CSS) was also discussed. GP5 (HR: 3.29, 95% CI 1.95–5.57; *p* < 0.001) and LDH ≥ 300 (HR: 7.82, 95% CI 3.71–16.48; *p* < 0.001) were risk factors in univariate analysis (Appendix A). On the other hand, BSI ≥ 1.5 was not a risk factor (HR: 1.61, 95% CI 0.86–3.04; *p =* 0.14). Multivariate analysis for CSS showed that only LDH ≥ 300 was an independent risk factor (HR: 10.14, 95% CI 2.36–43.61; *p =* 0.002).

### 3.3. Development of a Risk Model for Overall Survival

Patients with mHNPC were then classified into three groups according to three risk factors associated with OS. Accordingly, the low-risk group was defined as having none of the factors, the intermediate-risk group as those having one factor, and the high-risk group as those having two or three factors. The Kaplan–Meier cumulative OS is presented in Figure 3. The median OS was 162.0, 85.2, and 36.7 months in the low-risk, intermediate-risk, and high-risk group, respectively. Our findings showed that OS tended to decrease as risk increased (Log-rank test trend, *p* = 0.0005). Additionally, we have created a nomogram to predict 5-year survival using these three identified items (Appendix A).

## 4. Discussion

In the current study, the median OS of patients classified as high-risk based on the LATITUDE criteria was 55.06 months, which was similar to that in patients who received upfront abiraterone acetate and prednisolone in the LATITUDE study (53.3 months), despite most of the patients opting for CAB as their initial treatment [11]. Moreover, patients who were classified as high-volume according to the CHAARTED criteria had an OS of 52.93 months, which was longer than that in patients who received upfront docetaxel in the high-volume arm of the CHAARTED trial (49.2 months) [10]. This may be attributed to the increased sensitivity of Asian patients with metastatic PC to castration and their significantly longer survival times compared to other ethnic groups [19]. However, it is clear whether a certain number of patients with high-volume metastases respond well to ADT. Administering ARAT, such as abiraterone acetate, or docetaxel, to such patients may cause a decrease in the patients’ quality of life due to side effects, while increasing the burden of medical costs. For abiraterone, the incidence of grade 3 or 4 adverse events was 63% in the abiraterone group, compared to 48% in the placebo group [20]. We believe that this difference in incidence is not optimistic and is an impediment to the patient’s quality of life. Although the CHAARTED trial did not compare docetaxel with ADT alone in terms of the incidence of adverse events [10], the fatal febrile neutropenia associated with docetaxel is a significant reduction in the quality of life of patients.

Therefore, with the emergence of various treatment options, creating a new prognostic model, identifying patients who do not respond well to ADT, and opting for upfront ARAT or chemotherapy would be very meaningful.

Various studies have been available on the prognostic factors for mHNPC. Glass et al. suggested a prognostic model, which differentiated patients into three prognosis groups based on bone metastasis localization, performance status, PSA, and GS [21]. Cooperberg et al. proposed the J-CAPRA score as a prognostic model for progression. In this model, GS, PSA, and TNM stage, are scored as factors, with difference in progression-free survival observed between the three groups of patients classified according to their model [22]. Our multivariable analysis found that the presence of GP5, BSI ≥ 1.5, and LDH ≥ 300 IU/L, were prognostic factors for mHNPC. In addition, creating a prognostic model based on these three factors and classifying the patients into three groups (low, intermediate, and high risk) resulted in differences in OS among such groups (165.0, 85.2, and 36.7 months). Roy’s nomogram’s composition, including LDH, bone metastases, and Gleason score, may justify our results [15]. Our identification of GP5 and their scoring item GS9-10 are strongly related and consistent. Additionally, in their nomogram, the number of skeletal lesions is one of the predictors. We believe that our predictor, BSI, is an ideal item for a more objective assessment of bone involvement.

Several reports have suggested that the inclusion of GP5 increased the aggressiveness of PC. Kryvenko et al. reported that the presence of GP5 significantly increased the risk of metastasis, prostate cancer-specific survival (PCSS), and death [23]. Tsao et al. reported that patients with GS 9–10 tended to have a greater risk of metastasis and death after local treatment compared to patients with GS8 [24]. We also looked at the International Society of Urologic Pathology (ISUP) grade groups, and grade groups 4 and 5 (GS 8–10) were also poor prognostic factors for OS in univariate analysis (HR: 3.01, 95% CI 1.43–6.34; *p* = 0.004), but they were not significant in multivariate analysis (data not shown). Huynh et al. reported that the risk of OS was significantly higher for PC with GS3+5/5+3 than for that with GS4+4 [25]. Therefore, among ISUP grade group 4 (GS 8), there is a report that the inclusion of Gleason pattern 5 affects OS, which we believe supports our results. Including the presence of GP5 as one of the prognostic factors is appropriate.

Our study identified BSI ≥ 1.5 as one of the prognostic factors for mHNPC. The LATITUDE criteria indicated the presence of three or more bone lesions as a high-risk factor [11]. In the CHAARTED trial, ≥4 bone lesions with ≥1 beyond the vertebral bodies and pelvis were listed as a high-volume factor [10]. In previous reports, some prognostic models have been created based on the extent of bone disease (EOD) score [18]. Shiota et al. identified EOD 4 as a risk factor for metastatic high burden for PC [26]. Akamatsu et al. listed EOD 3 or higher as a prognostic factor for OS and reported that classifying patients into three risk groups resulted in significant differences [27]. We also divided the patients into two groups with low and high EOD score and investigated their OS. However, there were no significant differences between the two groups. In our study, EOD score was not a useful prognostic factor. EOD score is based on the number of bone lesions and do not account for the size of a single bone lesion. Moreover, the lesion count may contain some subjective factors. The BSI was developed as a marker of the total amount of bone metastasis using whole-body scintigraphy with 99 mTc-MDP [28,29]. One of the major features of the BSI is that the bone scintigram can be used to objectively evaluate the degree of bone metastasis throughout the body. Poulsem et al. reported that patients with PC and metastases of BSI ≥ 1.0 have an increased risk of PCSS than those with BSI < 1.0 [30]. One study showed that BSI > 3.5% was a significant determinant of death in the mHNPC group and that patients with a good BSI response to treatment (>45%) had lower mortality rates than those without such a response [31]. Suzuki et al. calculated the BSI of bone lesions beyond the vertebral bodies and pelvis (bBSI) and reported that patients with PC and bone metastases of bBSI > 0.27 had a significantly shorter OS [32]. We suggest using BSI to objectively quantify the degree of bone metastasis in order to establish a more accurate prognostic model.

Another prognostic factor, serum LDH, was found to be associated with OS in patients with mHNPC. LDH is an intracellular enzyme that is widely distributed in body tissues. When one of the tissues is injured and LDH is released into the blood, the serum LDH concentration increases. LDH plays an important role in cancer metabolism [33]. Notably, LDH has been reported as a prognostic factor for metastatic PC in several studies, with our results being consistent with these reports [34,35]. In the present study, visceral metastasis was not a significant prognostic factor for OS. Several reports have also shown that visceral metastasis is not a factor associated with OS [36]. In the current study, the percentage of patients with lung metastasis was high, whereas that of other visceral metastasis, was low. One study reported that among patients with high burden metastases, those with lung metastases had better OS than those with M1b [37]. Iwamoto et al. reported that patients with PC and lung metastases only had better OS than those with visceral metastases, except for lung metastases [38]. Racial differences may be one of the reasons why visceral metastasis was not a prognostic factor [39]. This should be investigated by accumulating more cases in the future. Several risk models have been reported for Japanese patients, and these are useful risk classifications that can be very clearly stratified [26,27]. However, these risk models are somewhat complicated to stratify, and the EOD score is included as a predictor of prognosis. Our model is very important in that it is simpler and more objective in predicting patient prognosis.

In the present cohort, most patients with mHNPC were treated with CAB. The 5-year survival rate for patients classified as low risk in our proposed risk model is approximately 80%. The prognosis for this group of patients is very favorable. In the CHAARTED trial, the 5-year survival rate for patients who received upfront docetaxel in the low volume setting was approximately 70% [10], indicating that low-risk patients in our risk model have good survival outcomes with CAB alone. In the ENZAMET trial, a subgroup analysis reported a 90% 3-year survival rate in low-volume patients treated with upfront enzalutamide [12]. Our study also shows that the 3-year survival rate for low-risk patients is comparable to the ENZAMET trial. In consideration of these factors, we believe that vintage therapy for low-risk patients can never be a substitute for chemotherapy or ARAT. Ideally, this prognostic model should be used to actually make treatment choices, but this will likely be a future challenge. A larger prospective cohort study will be needed to prove this.

The current study has several limitations worth noting. This study was retrospective in nature, and treatment selection and evaluation of the effects of treatment were left to the individual physicians, which may have resulted in bias. In addition, patients included herein were all Japanese. Thus, our results may not be applicable to other races. Given that only pretreatment factors were investigated in this study, we did not examine factors that may be predictive of post-treatment outcomes, such as response to initial treatment (PSA reduction rate, time to CRPC, etc.). In addition, the patients enrolled in the study ranged from 2000 to 2019. The addition of ARAT or chemotherapy as new treatment options over the past 20 years may be one limitation when considering survival, because before the coming of ARAT or chemotherapy, patients did not have the option of receiving that treatment. Moreover, there have been International Society of Urological Pathology Consensus Conference in 2005 and 2014, where revisions were made regarding the Gleason classification. Since this is a retrospective study and the data collection for pathology results is based on medical record entries, it is possible that shifting diagnoses and definitions over time may have resulted in bias. However, in this study, we are focusing on GP 5. Since there has been no significant revision in the ISUP Consensus Conference regarding the diagnosis and definition of GP5, we do not think that it has had a significant impact on this study. Finally, the choice of sequential treatment was also left to the discretion of the physician, and the change in survival rate due to the choice of treatment after castration-resistant PC had not been investigated.

## 5. Conclusions

We identified three prognostic factors for OS in patients with mHNPC: GP5 inclusion, BSI ≥ 1.5, and LDH ≥ 300. Using these three factors, we developed a new prognostic model for OS that can more objectively predict the prognosis of patients simply and objectively.

## Figures and Tables

**Figure 1 cancers-14-04822-f001:**
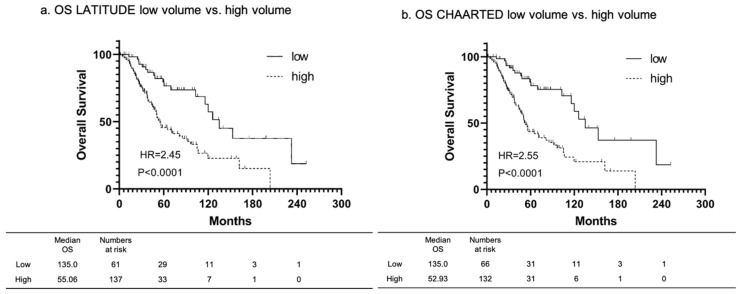
Kaplan-Meier showing the difference in OS classified by LATITUDE criteria, and CHAARTED criteria. (**a**) The high-risk group had a significantly shorter OS (HR: 2.45, 95% CI 1.48–4.00; *p* < 0.0001) than the low-risk group based on LATITUDE criteria. (**b**) The high-volume group had a significantly shorter OS (HR: 2.55, 95% CI 1.54–4.24; *p* < 0.0001) than the low-volume group based on the CHAARTED criteria.

**Figure 2 cancers-14-04822-f002:**
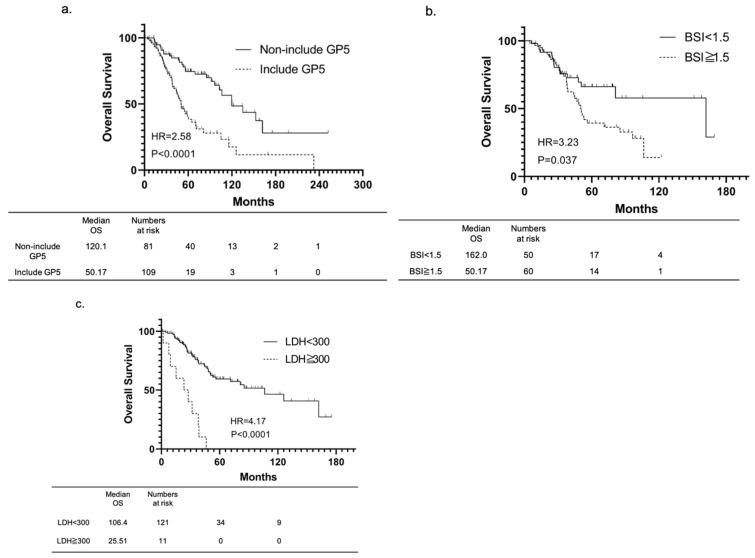
Kaplan-Meier showing the difference in OS stratified by inclusion of GP5, BSI ≧ 1.5, and LDH ≧ 300 IU/L. OS was significantly shorter with GP 5 inclusion (HR: 2.58, 95% CI 1.72–4.47; *p* < 0.001) than with GP 5 exclusion (**a**); in the BSI ≥ 1.5 group (HR: 3.23, 95% CI 1.77–5.89; *p* = 0.037) than in the BSI < 1.5 group (**b**); and in the LDH ≥ 300 IU/L group (HR: 4.17, 95% CI 2.10–8.30; *p* < 0.0001) than in the LDH < 300 IU/L group (**c**). Patients for whom information was available on each identified risk factors were selected. GP: Gleason pattern; BSI: bone scan index; LDH: lactate dehydrogenase.

**Figure 3 cancers-14-04822-f003:**
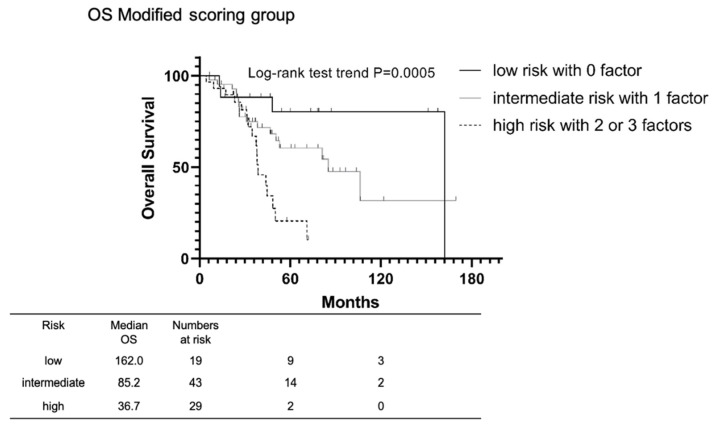
Kaplan-Meier estimates of cumulative OS in patients with mHNPC stratified by inclusion of GP 5, BSI ≧ 1.5, and LDH ≧ 300. Patients for whom information was available on all three identified risk factors were selected. Low risk patients had no risk factor, intermediate risk patients had one factor, and high-risk patients had two or three factors. OS tended to decrease as risk increased (Log-rank test trend, *p* = 0.0005). GP: Gleason pattern; BSI: bone scan index; LDH: lactate dehydrogenase.

**Table 1 cancers-14-04822-t001:** Patient characteristics.

Variables	Entire Cohort (*n* = 198)
Age, median (range)	71 (65–78)
*T stage, no (%)*	
T1-2	17 (8.6)
T3	85 (42.9)
T4	77 (38.9)
Unknown	19 (9.6)
*N stage, no (%)*	
N0	79 (39.9)
N1	115 (58.1)
Unknown	4 (2.0)
*M stage, no (%)*	
M1a	5 (2.5)
M1b	165 (83.3)
M1c	28 (14.1)
*Site of metastasis, no (%)*	
Lymph node	116 (58.6)
Bone	188 (94.9)
Lung	23 (11.6)
Liver	3 (1.5)
Adrenal gland	1 (0.5)
Initial PSA level, ng/mL, median (range)	230.5 (72.7–859.4)
*Gleason score, no (%)*	
≦3 + 4	10 (5.1)
4 + 3	19 (9.6)
8	52 (26.3)
≧9	107 (54.0)
Unknown	10 (5.1)
*Initial treatment, no (%)*	
CAB	185 (93.4)
LHRH agonist	3 (1.5)
Abiraterone	1 (0.5)
Other	5 (2.5)
Unknown	4 (2.0)

PSA: prostate specific antigen; CAB: combined androgen blockade.

**Table 2 cancers-14-04822-t002:** Univariate and multivariable analysis of prognostic factors for overall survival.

Variables	Univariate				Multivariable			
			95% CI				95% CI	
	*p* Value	HR	Lower	Upper	*p* Value	HR	Lower	Upper
Age <70 vs. ≧70 (years)	0.53	1.15	0.74	1.79	0.37	1.46	0.64	3.33
Include GP5	<0.001	2.78	1.72	4.47	0.045	2.77	1.03	7.49
PSA <200 vs. ≧200 (ng/mL)	0.40	1.21	0.78	1.86	0.32	0.55	0.17	1.78
BSI <1.5 vs. ≧1.5	0.04	1.91	1.03	3.53	0.033	3.48	1.10	11.00
Visceral metastasis	0.53	1.20	0.69	2.07	0.053	0.15	0.02	1.03
CRP <1.0 vs. ≧1.0 (mg/dL)	0.76	1.10	0.60	2.03	0.67	0.80	0.29	2.24
NLR <2.5 vs. ≧2.5	0.13	1.65	0.87	3.14	0.09	2.32	0.89	6.08
Hb <12 vs. ≧12 (g/dL)	0.35	1.35	0.72	2.56	0.64	0.79	0.29	2.15
ALP <300 vs. ≧300 (IU/L)	0.31	1.34	0.76	2.37	0.32	0.55	0.17	1.79
LDH <300 vs. ≧300 (IU/L)	<0.001	6.08	2.95	12.50	0.004	8.11	1.99	33.11

PSA: prostate specific antigen; GP: Gleason Pattern; BSI: bone scan index; CRP: C-reactive protein; NLR: neutrophil-to-lymphocyte ratio; Hb: hemoglobin; ALP: alkaline phosphatase; LDH: lactate dehydrogenase.

## Data Availability

The data presented in this study are available in the article and Appendix A.

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
