# Peer review of "Development of a Prognostic Model of Overall Survival for Metastatic Hormone-Naïve Prostate Cancer in Japanese Men"

_cancers, 2022, doi:10.3390/cancers14194822_

Round 1

Reviewer 1 Report

Dear authors!

A well done retrospective study.

I recommend publication after some minor changes:

1. I would change your title in: "Development of the prognostic model of overal survival for metastatic homone-naive prostatic cancer in japanese men"

2. please explain in more detail the "BSI" Score... i.e. "the bone scan index represents the percentage of total skeletal mass taken up by the tumor, is a reproducible quantitave expression of tumor burden seen on bone..."

3. Also please explain your chosen limit of BSI >1.5.

4. Did you perform a power calculation of your stats?

Apart from my minor recomendations well done, well written!

Reviewer 2 Report

This study retrospectively investigated prognostic factors to develop a predictive model for patients with mHNPC. The authors identified three prognostic factors, and concluded that GP5 inclusion, BSI ≥1.5, and LDH ≥300 were associated with OS. Creating a new prognostic model identifying patients who do not respond well to ADT, and opting for upfront ARAT or chemotherapy would be very meaningful is meaningful with the emergence of various treatment options. This prognostic model should be used to actually make treatment choices, this will likely be a future study.

Minor comment

It is suggested that a description of treatments for mHNPCs in the world and in your country at the time the subject was collected.

Reviewer 3 Report

The authors described a prognostic nomogram in metastatic castrate sensitive prostate cancer but the article at its present form is not acceptable. In the introduction, the authors should have cited the only nomogram that has been built for de novo metastatic castrate sensitive prostate cancer patients treated with ADT and modern ARAT combination (PMID: 35790787). This should have also been brought to the discussion. This could also help them justify the reason in choosing the predictors. The authors don't describe the nomogram, neither do they show any graphic for the nomogram which is concerning. Did the authors choose backward selection method for the final model. If that is the case, the model is never acceptable as backward selection is always flawed with overfitting and other additional concerns (Harrell, F. E. (2001)). The authors also did not describe what constitutes their testing dataset clearly. Did they split the data? Did they use an external validation set?

Reviewer 4 Report

     1-    Title: The title reflects the content well. Just a suggestion to replace “the” with “a” in “Development of the Prognostic Model of Overall Survival for Metastatic Hormone-Naïve Prostate Cancer.” Otherwise, they can leave the title as it is.

2-      English: The manuscript could benefit from editing for grammar, missing words, and subject-verb agreement, etc. It is recommended that authors delete irrelevant "general" phrases and sentences, repeated and unneeded words. They should use short sentences. Also, some Introductory sentences are irrelevant or are not needed. There are also typos in the manuscript.

3-      Abbreviations: All abbreviations should be revised and defined at their first use.

4-      Abstract: “Abstract: Background: To identify prognostic factors in patients with metastatic hormone-naïve prostate cancer (mHNPC) and develop a prognostic model based on these factors.” This is an incomplete sentence, and it reflects the objectives not the background.

5-      Introduction: “available treatments for CRPC, including alternative ADT, androgen receptor axis targeted agents (ARAT), and chemotherapy” What about radiation therapy?

6-      Introduction: “with ADT promoted longer overall survival.” Change promoted to prompted.

7-      Materials and methods: Authors used gleason pattern for grading PCa. Currently, grade group is the established grading system in PCa. Did the authors consider giving GGs for the cases and using them as variables instead of gleason pattern?

8-      Materials and methods: this section is poorly written. It should include subheadings such as “patients selection”, “ethical considerations”, “collection of clinical data”, “statistical analyses”, etc. Inclusion and exclusion criteria are missing. Statistical analysis tests used are missing.

9-      Results: there is discrepancy in results. Authors mentioned 193 patients included in the materials and methods and 198 patients in the results.

10-  Results: “88 patients succumbing to their disease.” What is meant by succumbing?

11-  Results: this section also can be divided into subheadings.

12-  Results: do the patients have any other malignancy that might have affected the results?

13-  Results: How did authors select certain cut offs for the different variables in Table 2 (70 years for age, PSA 200, etc.?

14-  Results: in the survival curves, did authors adjust for potential confounding factors?

15-  Figures: All the figure legends can be revised as to be more informative of the images presented. Also, statistical tests used and meaning of asterix need to be added. Abbreviations used withing Tables and Figures should be defined as well in the legends at the end.

16-  Discussion: Authors should focus more on the main findings and avoid repeating results presentation in the discussion. Authors could also correlate their findings with what has been published in literature. Clinical relevance should be added.

Round 2

Reviewer 3 Report

In absence of a validation or testing, the reliability and precision of a nomogram remains uncertain. 

Reviewer 4 Report

Thanks.